# The Role of Forkhead Box Proteins in Acute Myeloid Leukemia

**DOI:** 10.3390/cancers11060865

**Published:** 2019-06-21

**Authors:** Carmelo Gurnari, Giulia Falconi, Eleonora De Bellis, Maria Teresa Voso, Emiliano Fabiani

**Affiliations:** 1Department of Biomedicine and Prevention, University of Rome Tor Vergata, 00133 Rome, Italy; carmelogurnari31@gmail.com (C.G.); giulia_0312@hotmail.it (G.F.); debellis.eleonora.1@gmail.com (E.D.B.); Voso@med.uniroma2.it (M.T.V.); 2Fondazione Santa Lucia, Laboratorio di Neuro-Oncoematologia, 00143 Roma, Italy

**Keywords:** forkhead box proteins, acute myeloid leukemia, chemoresistance

## Abstract

Forkhead box (FOX) proteins are a group of transcriptional factors implicated in different cellular functions such as differentiation, proliferation and senescence. A growing number of studies have focused on the relationship between FOX proteins and cancers, particularly hematological neoplasms such as acute myeloid leukemia (AML). FOX proteins are widely involved in AML biology, including leukemogenesis, relapse and drug sensitivity. Here we explore the role of FOX transcription factors in the major AML entities, according to “The 2016 revision to the World Health Organization classification of myeloid neoplasms and acute leukemia”, and in the context of the most recurrent gene mutations identified in this heterogeneous disease. Moreover, we report the new evidences about the role of FOX proteins in drug sensitivity, mechanisms of chemoresistance, and possible targeting for personalized therapies.

## 1. Introduction

Forkhead box (FOX) proteins are an extended group of transcriptional factors characterized by the presence of an evolutionary conserved DNA-binding domain (DBD) named “winged-helix” or “fork-head”. The family name “fork-head” derives by the first gene discovered in *Drosophila Melanogaster* (*forkhead*, fkh) by Weigel et al. [1] in 1989 and was inspired by the fork-headed appearance of the mutated insect embryos, whereas the “winged-helix” name of the characteristic DBD present in all family members was suggested by the butterfly-like appearance of its three-dimensional structure. The prototypical DBD consists of about one hundred amino acids that under physiological conditions give rise to three α-helices, three β-sheets and two ‘wing’ regions that flank the third β-sheet [2].

FOX family members are now categorized, on the basis of sequence homology, into nineteen subgroups, from FOXA to FOXS, and reach at today the number of at least 50 genes distributed almost on all human chromosomes, Figure 1 [3,4].

Transcription factors are sequence-specific DNA-binding proteins (DBP) that control the rate of transcription of genetic information from DNA to messenger RNA, by binding to a specific DNA sequences (promoters and/or enhancers). As transcription factors, FOX proteins are responsible for the fine-tuning of gene expression during all stages of embryonic development and are guardians of the homeostasis in adult tissues. FOX proteins have been reported as active regulators of several networks, the main of which are: development, differentiation, maintenance of multipotency, proliferation, metabolism, DNA repair, cell cycle progression, migration, senescence, survival and apoptosis [5,6,7,8,9,10,11,12,13]. Despite the high sequence conservation of the *forkhead* domain, FOX proteins may exert different roles in the fine regulation of downstream genes, acting as repressors or activators of gene expression [14].

The mechanisms of gene expression regulation controlled by FOX proteins are, in some cases, so intricate that some FOX proteins are themselves the target of other members of the same gene family, as shown by Karadedou et al. that described the mechanisms by which FOXO3A and FOXM1 antagonize the activity of one another by regulating the transcription of downstream target genes [15]. The fine regulation of gene expression performed by FOX proteins is not only due to the tissue and/or cell-specific expression, but is also due to the post-translational modifications that mainly include phosphorylation, acetylation, ubiquitylation and sumoylation [16,17]. Post-translational modifications play a central role in cellular localization and activity of FOX factors. Mainly, FOX proteins act as transcriptional regulators in the nucleus, while they are prevalently inactive in the cytoplasm where they are subjected to proteasomal degradation.

The ability of FOX proteins to contribute to the control of several fundamental signaling pathways and of all the aspects of development and cell fate allows this superfamily of transcription factors to be heavily implicated in cancer initiation and progression. Indeed, FOX factors have been shown to play a role as either oncogenes or tumour suppressors, as well as active regulators of cellular resistance to chemotherapy and actionable targets in cancer therapy.

Myeloid neoplasms are a complex and heterogeneous group of hematopoietic diseases characterized by uncontrolled proliferation and/or blockage of differentiation of abnormal myeloid progenitor cells, and variable prognosis. “The 2016 revision to the World Health Organization classification of myeloid neoplasms and acute leukemia” categorizes myeloid malignancies into five primary types: myeloproliferative neoplasms (MPN), myeloid/lymphoid neoplasms with eosinophilia and rearrangement of PDGFRA (platelet derived growth factor receptor alpha), PDGFRB (platelet derived growth factor receptor beta), or FGFR1 (fibroblast growth factor receptor 1), or with PCM1-JAK2 (pericentriolar material 1-Janus kinase 2), myelodysplastic/myeloproliferative neoplasms (MDS/MPN), myelodysplastic syndromes (MDS) and acute myeloid leukemia (AML) and related neoplasms [18].

Accumulating evidence suggests that FOX proteins are profoundly involved in the maintenance of multipotency of hematopoietic stem cells (HSC) and in critical mechanism driving aberrant self-renewal in preleukemic cells [19].

In this review, we try to highlight the crucial role that FOX transcription factors play in acute myeloid leukemia development and progression, their role as potential direct and/or indirect therapeutic targets and as biomarkers of drug response and/or resistance.

## 2. Current Classification of Acute Myeloid Leukemia

Acute myeloid leukemia (AML) is a heterogeneous group of clonal disorder of the hematopoietic compartment characterized by abnormal proliferation of undifferentiated myeloid progenitors, impaired hematopoiesis, bone marrow failure and variable response to therapy.

Although AML arises in bone marrow hematopoietic stem cells, it may involve other extramedullary sites as lymph nodes, brain, spinal cord, liver, spleen, testicles and other parts of the body.

AML is classified according to “The World Health Organization (WHO) Classification of Tumours of Haematopoietic and Lymphoid Tissues”, which was last updated in 2008 [20] and revised in the aforementioned update of 2016 [18]. The current classification includes: AML with recurrent genetic abnormalities, AML with myelodysplasia-related changes, therapy-related myeloid neoplasm, and AML not otherwise specified (NOS). Incidence and survival outcomes vary according to age: in childhood it is a rare disorder with 7 occurrences per million annually and a 5-year overall survival (OS) exceeding 60%; in the adult setting AML is the most common form of acute leukemia with an incidence of 4–5 per 100,000 person/years and has the shortest survival (24% of five-year OS) [21,22,23]. Despite progress in treatment and supportive therapies, outcome of high-risk adult AML remains dismal with only less than 20% of patients becoming long-term survivors [24]. In this light, the identification of new actionable targets for the therapy of these diseases must be considered as a priority.

FOX proteins deregulation has been found closely related to several aspects of leukemia development, progression and therapy resistance.

## 3. Forkhead Box Proteins in RUNX1-RUNX1T1 Acute Myeloid Leukemia

One of the most frequent initiating alterations in AML is the AML1-ETO translocation t(8;21), accounting for about 10% of total AML [25]. Although, according to “The 2016 revision to the World Health Organization classification of myeloid neoplasms and acute leukemia” the t(8;21)(q22;q22.1), RUNX1-RUNX1T1 represents a specific subgroup of “AML with recurrent genetic abnormalities”, several authors have shown that the expression of RUNX1-RUNXT1 transcript in human hematopoietic stem progenitor cells (HSPC) causes deregulated differentiation and increased self-renewal of CD34+ cells without inducing AML [19,26]. Although, FOXO genes (FOXO1, FOXO3, FOXO4 and FOXO6) are frequently reported as tumour suppressors in several cancers [27,28,29,30,31,32,33,34,35], Lin et al. recently highlighted a new role of FOXO1 as an oncogene, clearly showing that the role of Forkhead box proteins depends on cellular context [19]. In this line, the Authors showed that the up-regulation of FOXO1 is required to sustain the growth of RUNX1-RUNXT1 cells, promoting the self-renewal and inhibiting the differentiation of human CD34+ HSPCs (Table 1). In particular, in RUNX1-RUNX1T1 CD34+ cells, increased levels of FOXO1 promote preleukemia transition and clonogenicity. Moreover, Lin et al. suggested the genetic and pharmacological ablation of FOXO1 as a therapeutic strategy for the elimination of preleukemic and leukemic t(8:21) HSPCs [19]. Indeed, an interesting therapeutic approach proposed for Acute Leukemia has been the interference with FOXO1 subcellular localization to improve blast cell sensitivity to antineoplastic drugs. Phosphorylation is the main cause of the 14-3-3 protein-mediated export of FOXO1 from an active nuclear form into the cytoplasm, where it results inactive [36,37]. Besides post-translational modifications, miRNA are emerging actors in regulating FOXOs levels and consequently AML blasts characteristics, such as drug sensitivity [38]. Together with FOXO3 activation, FOXO1 overexpression via miRNA interactions is involved in upregulation of ABCB1 gene, coding for P-glycoprotein, well known responsible for decreased drug accumulation in multidrug-resistant cancer cells [39].

## 4. Forkhead Box Proteins in PML/RARA (Promyelocytic Leukemia/Retinoic Acid Receptor-Alpha) Acute Promyelocytic Leukemia

Acute promyelocytic leukemia (APL) is another subgroup of acute myeloid leukemia characterized by a unique t(15;17) translocation generating the PML/RARA fusion gene.

The PML/RARA oncoprotein synthesis is the key pathogenetic event of APL specifically targeted by all-trans retinoic acid (ATRA) and arsenic trioxide (ATO), two non-chemotherapeutic agents that synergistically act inducing oncoprotein degradation. Although it is well known that this fusion protein blocks granulocytic differentiation by direct transcriptional inhibition of retinoic acid target genes [50], the detailed mechanisms and the complete list of genes involved in APL transformation are not fully understood. Although, the ATRA-related apoptosis of APL blasts via tumor necrosis factor-related apoptosis-inducing ligand (TRAIL) expression has been previously reported, the pivotal transcription factor has not been yet identified [51]. Sakoe et al. have shown that activation of FOXO3A is an essential event for ATRA-induced cellular response in human t(15:17) cell line NB4 [40]. Nuclear FOXO3A tunes HSC maintenance and, if phosphorylated, it is exported to the cytoplasm and interacts with 14-3-3 proteins, hence losing its function [52]. The phosphorylation of FOXO3A, and its consequent loss of function, is regulated by AKT pathway aberrantly activated in case of AML driver mutations such as FLT3-ITD or BCR-ABL [53], (Figure 2). In the APL setting, Sakoe et al. showed that ATRA treatment was able to reduce FOXO3A phosphorylation and to induce the translocation of the transcription factor in the nucleus, where the protein leads to apoptosis trough TRAIL upregulation [40]. Notably, FOXO3A silencing through shRNA inhibited ATRA-induced response in NB4 cells, as such as in ATRA resistant NB4/RA cells. ATRA treatment was unable to induce FOXO3A phosphorylation, TRAIL upregulation, apoptosis and granulocytic differentiation; whereas forced expression of active FOXO3A in the nucleus induced TRAIL production and apoptosis in NB4/RA cells [40] (Table 1). Thus, in APL, a member of FOXOs family acts as tumor suppressor gene differently from the previously reported role of FOXO1 in RUNX1-RUNX1T1 acute myeloid leukemia. These results highlight a possible role for FOXO3A as a potential therapeutic target in APL to overcome ATRA resistance.

Arsenic trioxide in combination with ATRA is currently considered the standard of care for adults with low-to-intermediate-risk APL, with a complete remission rate near to 100% [54]. Interestingly, Zhang et al. have recently shown that ATO treatment of gastric cancer cells induced the upregulation of FOXO3A expression in the nucleus and that FOXO3A knockdown attenuated the effect of ATO treatment in gastric cancer cells and in mouse models [55]. Moreover, they also demonstrate that ATO-related nuclear upregulation of active FOXO3A is the result of its phosphorylation via the aforementioned AKT pathway, leading to inhibition of cell migration. This new insight may be useful for further investigations about the role of FOXO3A in treatment response to ATO in APL patients. In this line, additional studies to identify novel therapeutic agents enabled to restore FOXO3A function may overcome ATO resistance even in patients harboring PML-A216V mutation, accounting for about 30% of ATO-resistant cases [56].

FOXO subfamily members are not the solely forkhead transcription factors involved in APL pathogenesis. In 2015, Somerville et al. reported the overexpression of FOXC1 in nearly 20% of primary non-APL AML samples, showing its involvement in the monocyte/macrophage differentiation block and in the increased clonogenic potential of AML cells [41]. FOXC1 is deregulated in different types of cancers and is often associated with poor prognosis in AML, cooperating with HOXA/B and consequently repressing the monocyte transcriptional regulator KLF4 [41]. More recently, our group showed that FOXC1 mRNA and protein levels were significantly lower in primary marrow samples from APL patients, as compared to samples obtained from patients with other AML subtypes, and normal CD34+ hematopoietic cells [42]. Moreover, we demonstrated that FOXC1 expression was significantly increased in APL samples following consolidation treatment and that ATRA treatment unlocked FOXC1 expression in NB4, but not in NB4-R4 ATRA-resistant cells. Of note, using chromatin immune precipitation assay (ChIP), we identified functional binding sites of ATRA in the FOXC1 promoter region. In addition, we showed that in diagnostic APL samples and in NB4 cells, reduced FOXC1 expression was associated to DNA hypermethylation of the +354 to +568 FOXC1 region and that hypomethylating treatment with decitabine of NB4 upregulated FOXC1 expression [42] (Table 1). Our findings indicate a dual repression model of FOXC1 expression in APL giving the rationale for a potential role of hypomethylating treatment (HMT) in advanced and/or resistant APL. HMT is anecdotally reported for the treatment of resistant APL cases and the rationale for their use has been postulated by the study of Soncini et al. where they show TRAIL-dependent apoptosis of APL and AML blasts [57]. In this work Soncini et al. identify a TRAIL promoter region, affected by DNA hypermethylation and whose function is restored after decitabine administration, demonstrating in APL a novel therapeutic approach other than PML-RARA degradation (ATRA-ATO scheme), using epigenetic drugs [57]. This strategy may be helpful to overcome resistance to differentiating treatment in case of mutations in PML or RARA.

## 5. Role of Forkhead Box Proteins According to Specific Gene Mutations Frequently Identified in Acute Myeloid Leukemia

Acute myeloid leukemia and related neoplasms are not only classified according to cytogenetic alterations, but also according to specific gene mutations, such as nucleophosmin 1 (NPM1), biallelic mutation of CCAAT/enhancer-binding protein alpha (CEBPA) and tunt-related transcription factor 1 (RUNX1) as provisional entity. Other recurrent mutations, such as FLT3-ITD (fms related tyrosine kinase 3-internal tandem duplication), TP53 (tumor protein p53), JAK2 (Janus kinase 2), K-RAS (Kirsten rat sarcoma viral oncogene homolog), N-RAS (neuroblastoma RAS viral oncogene homolog), ASXL1 (additional sex combs like 1, transcriptional regulator), TET2 (tet methylcytosine dioxygenase 2), DNMT3A (DNA methyltransferase 3 alpha), IDH1 (isocitrate dehydrogenase type 1), IDH2 (isocitrate dehydrogenase type 2), SRSF2 (serine and arginine rich splicing factor 2), SF3B1 (splicing factor 3b subunit 1), U2AF1 (U2 small nuclear RNA auxiliary factor 1), etc, [58] do not define, at today, a specific genetic subgroup of acute myeloid leukemia since they are spread across most subgroups, but clearly have prognostic importance.

New insights show that many FOX proteins are downstream targets of some recurrent somatic mutations in myeloid neoplasm. In this line, future studies may be helpful to identify FOX proteins as predictive marker of survival, early biomarkers of response and actionable targets for AML therapy. Here we report about the interaction between FOX proteins and FLT3-ITD, NPM1 and IDHs somatic mutations in AML.

## 6. Forkhead Box Proteins in Nucleophosmin (NPM1) Mutated Acute Myeloid Leukemia

Leukemias with mutated NPM1 have been recognized as a specific subgroup of AML [18]. NPM1 is a nuclear-cytoplasmic shuttle protein belonging to the nucleophosmin/nucleoplasmin family of chaperones and is mutated in approximately 35% of de novo AML cases [59]. Mutations of the NPM1 gene cause abnormal subcellular localization (cytoplasmic) of this nuclear shuttle-protein and are drivers of a typical subset of de novo AML associated with normal karyotype, FLT-3 mutation, specific morphological features and better prognosis in the majority of cases [60]. Falini et al. demonstrated a different gene expression profile of this AML subset characterized by up-regulation of genes involved in stem-cell maintenance and in 2011 Bhat et al. identified a region responsible for the interaction of NPM1 with FOXM1 [59,61]. FOXM1 is a transcription factor involved in the execution of the mitotic program and is overexpressed in proliferating cancer cells but not in quiescent ones as well as in terminally differentiated cells [62]. Different pathways like Ras-MAPK, Sonic Hedgehog and NF-kB cause activation of FOXM1 while p53 leads to its downregulation, thus acting as a potential oncogene required for tumor growth and spread, being defined the “Achilles’ heel” of cancer [63]. As a matter of fact, FOXM1 is overexpressed in multiple types of cancer, similarly to NPM1, being upregulated in highly dividing cancer cells and associated with genomic instability and aneuploidy signatures, although the latter is not the case of NPM1 mutated AML whose one of the main feature is harboring a normal karyotype [64,65]. NPM1 via its heterodimerization domain binds to the transactivation domain of FOXM1 and its knockdown leads to downregulation of FOXM1 expression. Moreover, in OCI/AML3 leukemia cells, where mutant NPM1 is localized in the cytoplasm [66], it has been demonstrated that typically nuclear FOXM1 is predominantly co-localized with NPM1 in the cytoplasm, while NPM1 knockdown leads to the disappearance of FOXM1 from the cytoplasm, suggesting that NPM1 may also determine intracellular localization of FOXM1 [67]. In addition, it has been shown that the cytoplasmic relocalization and subsequent inactivation of FOXM1 is probably responsible of the favourable outcome of NPM1 mutated AML [45]. The stable knockdown of FOXM1 in AML cells lines KG-1 and MV4-11 results in increased sensitivity to cytarabine, the backbone for the initial treatment of AMLs together with an anthracycline. Thus the overexpression of FOXM1 may confer chemoresistance (Table 1, Figure 3). Targeting the interaction between FOXM1 and NPM1 or downregulating the FOXM1 by peptides or small molecules may represent a novel anti-leukemic strategy. Recently some data indicate that proteasome inhibitors suppress FOXM1 by inducing HSP70 [68]. Clinical trials have demonstrated their feasibility in combination with conventional chemotherapy for the treatment of resistant AML and acute lymphoblastic leukemia (ALL) with neurological toxicity as the most common adverse event [69]. Central and mostly peripheral neurotoxicity has been reported in case of bortezomib—the first in class proteasome inhibitor approved for the treatment of multiple myeloma—administration, resolved or controlled using neurotrophic drugs, gabapentin or neurophysiotherapy [70]. In summary, FOXM1 may represent a novel therapeutic target to overcome NPM1 refractory/resistant AMLs, a little group who may take advantage of adding these new drugs to the conventional rescue strategy still lacking a high efficacy.

## 7. Forkhead Box Proteins in FLT3-ITD Mutated Acute Myeloid Leukemia

Fms-related tyrosine kinase 3 internal tandem duplication (FLT3-ITD) gene mutations were first reported in 1996 by Nakao et al. [71] and were observed in about 20–30% of AML patients. FLT3-ITD mutation confers a poor prognostic value with a high leukemic burden and shorter OS if compared with patients without this mutation [72,73]. According to 2017 European Leukemia Net guidelines, mutant-to-wild-type allelic ratio and the presence of concomitant NPM1 mutation influence the prognosis of this setting of patients [74]. FLT3-ITD receptors exhibit constitutive tyrosine kinase activity leading to suppression of apoptosis and increased cell division. Scheijen et al. using a model system for assessing potency and downstream signaling of kinase oncogenes (Ba/F3 cells) demonstrated that FLT3-ITD expression results in activation of Akt and concomitant phosphorylation of the FOXO3A, required for their translocation from the nucleus into the cytoplasm [46]. The Authors also shown that FLT3-ITD expression prevents FOXO3A-mediated apoptosis, promoting cells survival and proliferation, suggesting an oncosuppressive role of FOXO3A in this setting of AML [46] (Table 1). On the other hand, Seedhouse et al. identified, in a large cohort of trial samples, a lower FOXO1 expression level in FLT3-ITDs AML cells (*p* < 0.001), whereas they do not find statistically significant differences in FOXO3 and FOXO4 expression [47]. The Authors also reported about the reactivation of FOXO1 expression through FLT3-ITD inhibition using siRNA assays, defining the involvement of this gene in poor risk group of AML [47]. Thus, further studies are needed to better clarify the effect of the altered mRNA expression of FOXOs genes and their subcellular localization in FLT3-ITD positive AML. In addition to its role in APL setting, it has been shown that upregulation of FOXC1 has a poor prognostic role in FLT3-ITD AML [41,75]. In a series of 765 FLT3-ITD patients the high expression of FOXC1 was significantly related to chemoresistance and high cumulative incidence of relapse. Cauchy et al. indeed, demonstrated that FLT3-ITD mutation leads to upregulation of multiple pathways as MAPK, NF-κβ and PI3k promoting cell proliferation [76,77]. However, the molecular mechanism is not completely understood and future studies are warranted to clarify it and to assess the utility of FOXC1 as a potential cancer biomarker of induction chemotherapy outcome to improve risk-stratification of AML.

Beyond FOXC1 and FOXO subfamily members, FOXM1 expression levels were found deregulated in FLT3-ITD cell lines and patients. In particular, Liu et al. showed that expression of FOXM1 in AML patients was correlated with the presence of FLT3-ITD and overall survival [48]. They also used an in vitro model of FLT3-ITD positive cell line (MV4-11) and THP1 FLT3-ITD negative cells to test FOXM1 expression in response to FLT3 receptor tyrosine kinase inhibitor AC220 (quizartinib) or FLT3 ligand (FL) and showed that inhibition of FLT3-ITD by AC220 down-regulated FOXM1 expression in MV4–11 cells, while the stimulation of FLT3 by FL up-regulated FOXM1 expression in both MV4–11 and THP1 cells. Moreover, the addition of the FOXM1 inhibitor thiostrepton (TST) to FLT3-ITD positive and negative AML cells was able to induce the apoptosis of MV4–11 and THP1 in a dose-dependent manner, concluding that FOXM1 may be useful as potential prognostic marker and therapeutic target in AML [48]. More recently, Khan et al. performing a multi-institution retrospective study, were able to link FOXM1 expression to clinical outcomes in AML [45]. They found FOXM1 expression level as an independent clinical predictor of chemotherapeutic resistance in intermediate-risk AML and showed that FOXM1 play an important role in the clonogenic activity of AML cells [45]. In particular, using colony assays, they showed that FOXM1 knockdown in AML cell lines induced a dramatic decrease in both colony size and number. On the contrary, using transgenic mouse models, they showed that constitutive overexpression of FOXM1 induced chemoresistance, suggesting FOXM1 as a critical mediator in the emergence of resistant leukemic clones [45]. Of note, the addition of a proteasome inhibitor, ixazomib, was useful to increase the sensitization of AML cells to both cytarabine and the hypomethylating agent 5-azacitidine [45]. All in all, the previously mentioned works make clear evidence on the role of FOXM1 as a co-modulator of the AML progression and treatment response, suggesting that the targeting of FOXM1 may be useful, in combination with standard therapy, in AML treatment. Moreover, further studies are needed to unravel the role of FLT3-inhibitors and FOXs protein family members in order to identify possible markers of drug response as well as novel therapeutic targets.

## 8. Forkhead Box Proteins in IDHs Mutated Acute Myeloid Leukaemia

Isocitrate dehydrogenases (IDHs) are enzymes mainly involved in epigenetic regulation. The term epigenetic refers to a set of heritable phenotype changes that do not involve alterations in the DNA sequence, but can switch genes on or off and consequently determine which proteins will be transcribed and which others will be silenced, respectively. The connection between epigenetic changes and the occurrence and development of tumors has been extensively studied in the last decades [78,79]. In particular, DNA methylation is a chemical process that adds a methyl group to specific DNA sequences, frequently CpGs dinucleotide, modulating the gene expression. This process is catalyzed by epigenetic regulatory enzymes, including DNA methyltransferases, methylcytosine dioxygenases and isocitrate dehydrogenases. Epigenetic regulatory enzymes have been recognized as mutated in various types of cancer, and mutations of these enzymes have been closely related to the malignant phenotype [80,81,82,83,84].

In particular, isocitrate dehydrogenases 1 (IDH1) and isocitrate dehydrogenases 2 (IDH2), the cytoplasmic and the mitochondrial isoform, respectively are enzymes of Krebs cycle, converting isocitrate, generated in the tricarboxylic acid cycle (TCA), to alpha-ketoglutarate (α-KG) leading to restoration of cellular NADPH (Nicotinamide adenine dinucleotide phosphate-H) [85]. Mutations in IDH1 and IDH2 are found in different cancer types including brain tumor (over 70% of gliomas), colorectal cancer, prostate cancer and hematological malignancies as AML and myelodysplastic syndromes [86,87,88]. The consequent alteration of enzymatic activity causes the production of D-2-hydroxyglutarate (2-HG) instead of α-KG, a putative oncometabolite that interferes with mitochondrial function, imbalances the cellular redox potential and inhibits the dioxygenase as TET enzymes, resulting in epigenetic deregulation [88,89].

Discovered approximately 10 years ago by Marcucci et al. [90], hot spot mutations in IDH1 (R132) and IDH2 (R140 and R172) occur in approximately 20–30% of patients with cytogenetically normal AML, 7–14% for IDH1 and 8–19% for IDH2 clustering in the region of isocitrate binding and with an uncertain impact on survival outcomes [90,91,92]. IDH1 mutations prevalently occur in the same hotspot (R132) and their prognostic impact is strictly dependent on co-occurent mutational status, being a large part of IDH1 mutated AMLs also NPM1 mutated. IDH1 mutations are targeted by ivosidenib, a first-in-class IDH1 inhibitor [93].

IDH2-R140 mutations occur more frequently than IDH2-R172 mutations together with other epigenetic somatic mutations, identifying two distinct subgroups of patients with different prognosis (the former with better prognosis) and mutational status. These mutations are targeted by enasidenib, which, like ivosidenib, may cause a differentiation syndrome with a clinical picture similar to that caused by ATRA in APL [94,95].

IDH1 is directly regulated by FOXO transcription factors and, in case of mutations, this regulation leads to the upregulation of 2-hydroxyglutarate (2-HG). Charitou et al. firstly discovered the dual function of FOXOs in mediating both tumor suppression and promotion in the case of IDH1 mutation [49]. In wild-type cells it has been shown that FOXOs regulate IDH1 and, as a consequence, cellular differentiation and tumor suppression, probably ensuring adequate NADPH, α-KG and GSH cytoplasmic levels, providing protection against genomic instability and oxidative stress. In IDH1 mutated cells the increased levels of 2-HG produce inhibition of dioxygenase function, epigenetic instability and tumor progression [49] (Table 1, Figure 4).

Recently, the two novel aforementioned agents selective for these mutations have been approved by the FDA (Food and Drug Administration), ivosidenib for IDH1 and enasidenib for IDH2 mutations [96]. Other IDH1/2 inhibitors, such as vorasidenib (AG881) a panIDH inhibitor, are now under investigation [97]. Moreover, clinical trials with the BCL2 inhibitor venetoclax, already used in chronic lymphocytic leukemia, have shown promising results in AML patients in particular in IDHs mutated cases [98,99]. The discovering of FOXOs function may lead to novel targeted therapies to restore FOXO as tumor suppressor.

## 9. Forkhead Box Proteins in Refractory Acute Myeloid Leukemia: Mechanism of Chemoresistance and Treatment Failure

One of the first genes belonging to the FOX superfamily that was associated with the pathogenesis of AML is FOXM1. Already mentioned in the case of NPM1 and FLT3 mutated AML, it stimulates expression of cyclins as CCNA2 and CCNB1 and regulates different pathways involved in cell proliferation and division [43]. The importance of FOXM1 in AML is underlined by the evidence that its reduction results in inhibition of proliferation in AML blasts, through a decrease of the Aurora kinase B, Survivin, Cyclin B1 and an increase of p21 and p27, thus, altering the balance between apoptosis and cell survival [44] (Table 1). Furthermore, FOXM1 is a downstream effector of Ras-Raf-MEK-ERK pathway and TP53/Rb, both implicated in AML pathogenesis and development. FOXM1 is targeted by microRNAs (miRNAs) such as miR-370 and its expression plays a key role in AML initiation and progression [100]. In de novo AML, the expression level of miR-370 is significantly reduced compared to that of healthy controls and it has been shown that azacitidine may re-increase its level above normal range. Zhang et al. demonstrated that FOXM1 expression is 21-fold higher in de novo AML than in normal bone marrow cells and that it reduces with the achievement of marrow CR during treatment [100]. As aforementioned in FLT3 paragraph, Khan et al. investigated FOXM1 expression in intermediate-risk AML, a group including more than 50% of de novo AML cases, whose treatment approach and prognosis are really heterogeneous [45]. They studied 111 intermediate-risk AML showing that higher expression of FOXM1 was significantly related to treatment resistance especially to cytarabine-containing treatment. Proteasome inhibitors, such as ixazomib, have been shown to counteract FOXM1, stabilizing HSP70 hence restoring chemosensitivity [101]. Of note, Barger et al. have recently studied the role of FOXM1 variants (FOXM1a,b,c), beginning to dissect their complex role in the cellular context of cancer cells [65]. Nevertheless further studies are warranted to explore the function of FOXM1 variants in AML biology. In summary FOXM1 inhibition may be an attractive target for leukemia therapy and addition of FOXM1 inhibitors, as ixazomib, to standard regimens may be an attractive way to give a possibility to chemo-resistant patients as well as elderly one who cannot tolerate standard dosage of chemotherapy drugs.

FOXOs are one of the largest subgroups of forkhead family members, including FOXO1, FOXO3, FOXO4 and FOXO6, bona fide tumor suppressors. Their aberrant expression plays an important role in leukemogenesis. Moreover, FOXO3 is implicated in BCR/ABL fusion gene leukemogenesis, a translocation occurring in chronic myeloid leukemia, acute lymphoblastic leukemia and more rarely AML [102]. FOXO3A in particular regulates multiple oncogenic kinases such as Ras-Raf-MEK-ERK pathway, Akt and IkB kinase [103,104]. It is demonstrated that FOXO activity is reduced in AML with consequent activation of JUN/c-JUN pathway rendering the clone chemoresistant. FOXO3A, in particular, is inactivated in AML due to its cytoplasmic localization and degradation via MDM2 ubiquitination [105] and subsequent constitutive activation of inhibitor of nuclear factor kappa B kinase subunit beta/nuclear factor kβ (IKK/NFκB) is essential for AML blast cells. Thus, an increasing activity of FOXO3A, via the inhibition of IKK activity or Ras-MAPK pathway, may potentially result in an alternative therapeutic strategy for treating AML. Moreover, it has been recognized that FOXO3 inactivation is due to its highly phosphorylated state and levels of total pFOXO3A (phosphorylated FOXO3A) are higher at AML relapse if compared with diagnosis [28]. Higher levels of FOXO3a are already been correlated with poorer prognosis in AML with normal cytogenetics (NK-AML). In this setting of AML, FOXO3a level maintains its prognostic role even if related to the FLT3/NPM1 mutated status, being NPM1mut/FLT3 wt patients the lowest expressors [106]. Kornblau et al. investigated the expression and inactivation of FOXO3a trough phosphorylation in a large cohort of newly diagnosed AML patients [28]. Higher levels of pFOXO3a has an adverse prognostic impact on overall survival, being overexpressed in M5 if compared with M2 AML subtype as well as in FLT3-ITD mutated AMLs and in cases with higher WBC, marrow and peripheral blasts count. AML with these characteristics are characterized by primary resistance, shorter remission duration and inferior OS. Developing therapies to restore the phosphorylation of FOXO3a, its activation and normal subcellular localization may overcome primary resistance. Decitabine and, to a lesser degree, azacitidine indirectly induce dephosphorylation of FOXO3a and its re-translocation to the nucleus, restoring its function. This is probably due to activation of targets as PUMA (p53 upregulated modulator of apoptosis) and BIM (Bcl-2-like protein 11, commonly called BIM) and induction of apoptosis. Moreover, it is known that hypomethylating agents can inhibit NFκB pathway, whose importance has already been discussed [107].

FOXP1 is one of the FOXP subfamily members which, as the other forkhead box transcription factors, regulates cell differentiation and proliferation [14]. FOXP1 supports leukemic cell expansion, being upregulated by PUM-1and PUM-2, known to be important for hematopoietic stem cell renewal [108].

Finally, FOXP3, together with CD4 and CD25, is the marker of regulatory T cells (Tregs), a subset of T lymphocytes known to have a critical role in cancer development. In the setting of AML patients, it has been shown that FOXP3 polymorphisms in the donor may have a prognostic impact on clinical outcome of pediatric patients undergoing hematopoietic stem cell transplant (HSCT) in terms of higher incidence of veno-occlusive disease (VOD), Cytomegalovirus (CMV) infection and OS [109]. Moreover, the expression of FOXP3 decreases at the time of both acute and chronic graft-versus-host-disease (GVHD) onset in patients undergoing HSCT [110].

## 10. Future Perspectives

As summarized in this review, in the last decades, accumulating evidences have suggested wide implications of forkhead box proteins in AML pathogenesis, prognosis and treatment response, beginning to dissect the complex panoplia of their multifunctional and subcellular interactions. Nevertheless several points remain to be addressed.

First of all, as previously described, FOX family members have been identified as both tumor suppressors and oncogenes, depending on cell type and context. This ambiguous function urges to be better-unraveled trough future studies on well-characterized subgroups of AMLs.

Nowadays, the study of bone marrow microenvironment is emerging as an important actor in AML biology [111,112]. In particular, mesenchymal stem cells (MSCs) are important components of the hematopoietic microenvironment that support hematopoietic stem cell maintenance, self-renewal, and differentiation, through hematopoietic–stromal interactions, production and secretion of cytokines [113,114]. In our review of the literature, only a few authors have begun to investigate the putative role of FOXM1 protein in the cross-talk between bone marrow and microenvironment [11]. Thus, we strongly suggest to extend this studies to other members of FOX family proteins to better understand this aspect of AML biology as a future plan.

Furthermore, single nucleotide polymorphisms (SNP) were widely implicated in AML pathogenesis. SNPs have been identified as risk factors for AML development [115,116,117] and as prognostic factors in terms of survival outcomes and drug sensitivity/resistance [118,119,120,121]. FOXP3 SNPs have been recently identified as prognostic factors in pediatric and adult patients undergoing HSCT [122]. A comprehensive study of FOX family members SNPs may be another attractive and informative research field in AML.

Beyond the aforementioned function of DNA methylation in AML biology, miRNAs are emerging as important players in epigenetic modulation of gene expression. However, only few studies have tried to elucidate the interaction between miRNAs and FOX family members in AML pathophysiology.

Already mentioned in the case of APL, TRAIL and its receptors are often deregulated in AML resulting in drug-resistance, probably due to a deficiency of NF-κβ pathway via PU1 and upregulation of anti-apoptotic genes [123]. The discovery of new agents targeting TRAIL pathway may be an attractive field of research in the next years.

Moreover, understanding more about FOX protein-mediated mechanisms of chemosensitivity and resistance will address the unmet clinical need of patients unfit for standard treatment or resistant to conventional chemotherapy. The novel evidences about FOX proteins-related chemoresistance may lead to combine FOX protein regulators with lower doses of chemotherapeutic drugs extending the possibility of being treated also to unfit and elderly patients.

Thus, targeting these pathways may represent a new way of treating acute myeloid leukemia, moving towards a more specific and personalized medicine approach. The study of FOX proteins at different disease specific time points, such as diagnosis and relapse, will ameliorate the prognosis of patients with AML together with other well recognized prognostic markers like molecular biology and cytogenetics in order to better characterize the individual risk and have more information about the drug sensitivity profile.

## Figures and Tables

**Figure 1 cancers-11-00865-f001:**
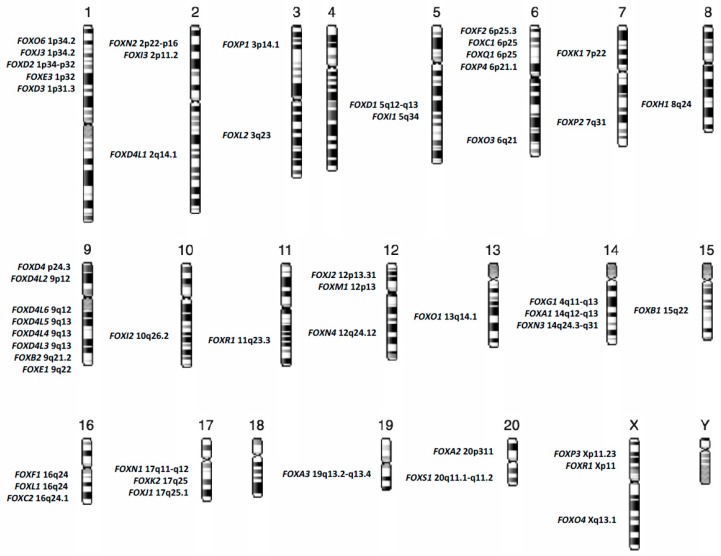
Chromosomal distribution of forkhead box (*FOX)* genes in the human genome.

**Figure 2 cancers-11-00865-f002:**
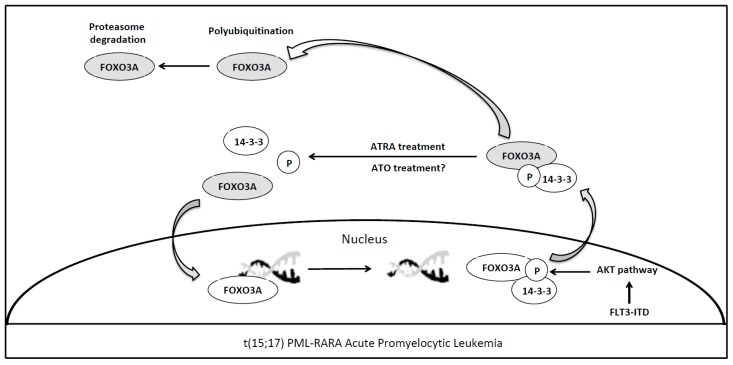
ATRA mediated reactivation of FOXO3A in t(15;17) PML-RARA acute promyelocytic leukemia. Nuclear FOXO3A (active form), once phosphorylated, interacting with 14-3-3 protein is exported to the cytoplasm losing its function [52]. The phosphorylation of FOXO3A, and its consequent loss of function, may be regulated by AKT pathway aberrantly activated in case of AML driver mutations such as FLT3-ITD [53]. All-trans retinoic acid (ATRA) treatment is able to reduce FOXO3A phosphorylation and to induce the relocation of the transcription factor into the nucleus, where the protein leads to blast apoptosis [40]. Arsenic trioxide (ATO) treatment in this cellular context needs to be better assessed. Light grey shows the FOXO3A inactive form.

**Figure 3 cancers-11-00865-f003:**
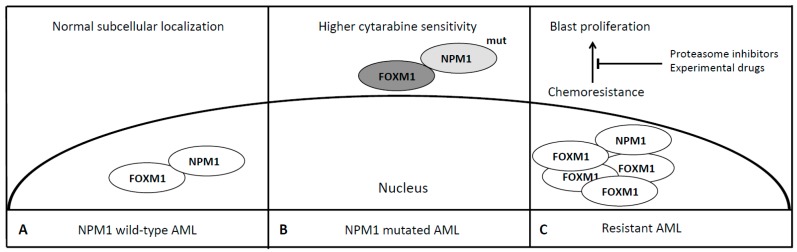
FOXM1 in NPM1 and refractory AMLs. In NPM1 wild-type AML, FOXM1 co-localizes with NPM1 in the nucleus in its active form (**Panel A**). In NPM1 mutated AML, the aberrant cytoplasmatic localization of NPM1 determines the cytoplasmatic trapping of FOXM1 [67] and its subsequent inactivation, probably responsible of the favourable outcome of NPM1 mutated AML [45] (**Panel B**). FOXM1 is overexpressed in case of chemoresistant AML [65]. Proteasome inhibitors and other experimental drugs may overcome drug resistance restoring FOXM1 normal levels [69] (**Panel C**). Dark grey, Inactive FOXM1; light grey, mutated NPM1.

**Figure 4 cancers-11-00865-f004:**
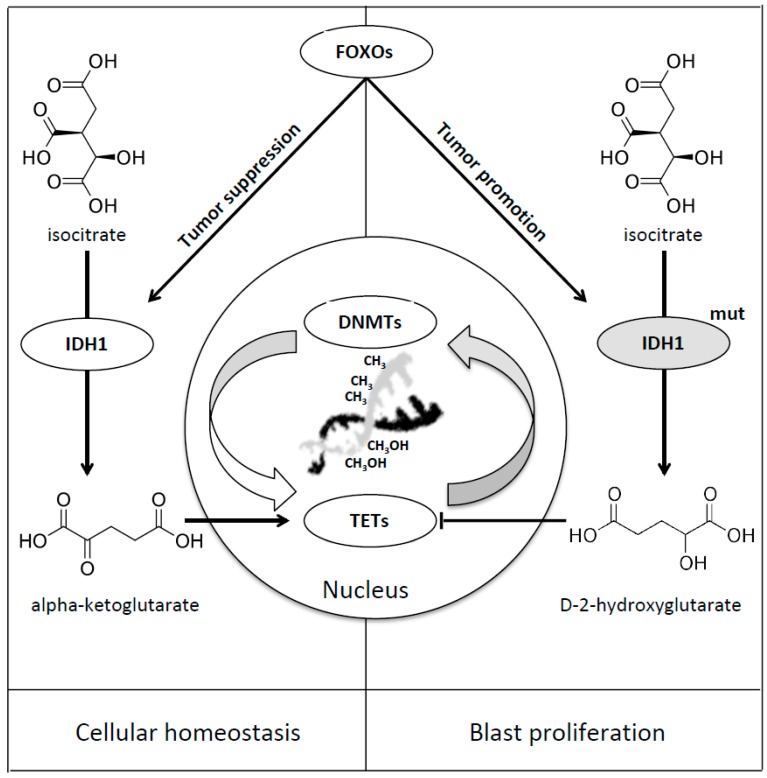
FOXOs mediated regulation of IDH1. IDH1 is directly regulated by FOXO transcription factors [49], taking part in the enzymatic reduction of isocitrate to alpha-ketoglutarate, ensuring cellular homeostasis, thus playing its tumor suppressive role (Left Panel). In case of IDH1 mutation FOXOs transcription factors enhance the function of IDH mutated enzyme and, as a result, lead to the accumulation of the D-2-hydroxyglutarate onco-metabolite and promote blast proliferation (Right Panel) [49].

**Table 1 cancers-11-00865-t001:** Forkhead box factors deregulation according to acute myeloid leukemia (AML) recurrent abnormalities and their involvement in biological processes.

Recurrent Abnormalities in Acute Myeloid Leukemia	FOX Family Member	Biological Process	References
t(8;21); RUNX1-RUNX1T1	FOXO1	Self-renewal and Differentiation	Lin et al. [19]
PML-RARA	FOXO3A FOXC1	Apoptosis and Granulocytic Differentiation Granulocytic Differentiation and Epigenetic Regulation	Sakoe et al. [40] Somerville et al. [41] and Fabiani et al. [42]
NPM1	FOXM1	Cell Proliferation, Division and Chemoresistance	Laoukili et al. [43] Nakamura et al. [44] Khan et al. [45]
FLT3 ITD	FOXO3A FOXO1 FOXM1	Apoptosis, Survival and Proliferation Cell growth, Apoptosis and antioxidant defences Survival, Apoptosis and Chemoresistance	Scheijen et al. [46] Seedhouse et al. [47] Liu et al. [48]
IDH 1-2	FOXOs	Cellular Differentiation and Tumor Suppression/Progression and Epigenetic Instability	Charitou et al. [49]

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
