# Peer review of "The Role of Forkhead Box Proteins in Acute Myeloid Leukemia"

_cancers, 2019, doi:10.3390/cancers11060865_

Round 1

Reviewer 1 Report

The authors describe in detail the role of the various forkhead transcription factors in AML classified by recurrent cytogenetic or molecular abnormalities. This is a relevant topic. However little detail or insight is provided as to the mechanism by which the transcription factor is conferring its biologic effect. The FOX transcription factors are oncogenes or tumor suppressors depending on cellular context. While the authors allude to this in the introduction it would be nice to see this discussed in more detail in the specific molecular subsets of AML described. What are the relevant targets in each subset of AML. Are there transcriptional targets or protein interactions that are critical to its biologic effect? What is the progress in developing targeted therapies against this family of transcription factors? This would make the therapeutic relevance of the paper more clear. A schematic figure would be helpful clarifying oncogenic and tumor suppressive roles of FOX transcription factors depending on disease context

Author Response

Reviewer 1

Comments and Suggestions for Authors

The authors describe in detail the role of the various forkhead transcription factors in AML classified by recurrent cytogenetic or molecular abnormalities. This is a relevant topic. However little detail or insight is provided as to the mechanism by which the transcription factor is conferring its biologic effect. The FOX transcription factors are oncogenes or tumor suppressors depending on cellular context. While the authors allude to this in the introduction it would be nice to see this discussed in more detail in the specific molecular subsets of AML described. What are the relevant targets in each subset of AML. Are there transcriptional targets or protein interactions that are critical to its biologic effect? What is the progress in developing targeted therapies against this family of transcription factors? This would make the therapeutic relevance of the paper more clear. A schematic figure would be helpful clarifying oncogenic and tumor suppressive roles of FOX transcription factors depending on disease context

Dear Reviewer,

thank you very much for your kind revision of our work. According to all of your suggestions, we have improved the explanation of the dual role of FOX proteins as both oncosuppressors and oncogenes, depending on subcellular localization and according to AML subtype.

Moreover, following the Editor suggestion, we have also extended several paragraphs and 3 figures, summarizing the role of FOX factors depending on disease context and subcellular localization, were added, too.

In this line, is not easy for us to report in this response letter all the changes made in the text as line-by-line. Nevertheless, for your usefulness, all changes are reported in bold. 

We hope our manuscript is now suitable for publication in “Cancers”.

Sincerely,

Emiliano Fabiani

Reviewer 2 Report

In this review, the authors provided a nice over view for the contribution of FOX transcription factors to AML pathogenesis. It is well written and well organized. I am in favor to publish. It would be a good addition if the authors can make a figure summarizing the key points described in the text.

Author Response

Reviewer 2

Comments and Suggestions for Authors

In this review, the authors provided a nice over view for the contribution of FOX transcription factors to AML pathogenesis. It is well written and well organized. I am in favor to publish. It would be a good addition if the authors can make a figure summarizing the key points described in the text. 

Dear Reviewer,

thank you very much for your kind revision of our work. According to your suggestion, we have added 3 figures to summarized the role of FOX proteins as both oncosuppressors and oncogenes, depending on subcellular localization and according to AML subtype.

We hope our manuscript is now suitable for publication in “Cancers”.

Sincerely,

Emiliano Fabiani

Reviewer 3 Report

In their manuscript “The role of Forkhead box proteins in Acute Myeloid Leukemia”, Gurnari et al. discuss the role of FOX transcription factors for drug sensitivity in acute myeloid leukaemia (AML). 

Their introduction to FOX genes and proteins is concise, and their review provides a relevant summary.

Minor comments:

Line 59-62: “FOX proteins have been reported as active regulators of several networks, the main of which are: development, differentiation, maintenance of multipotency, proliferation, metabolism, DNA repair, cell cycle progression, migration, senescence, survival and apoptosis.” Relevant references should be incorporated.

Line 77-79: dto.

Line 96-97: “Acute myeloid leukaemia (AML) is a heterogeneous group of clonal disorder of the haematopoietic stem cells”. This is misleading as AML does not (necessarily) arise from the HSC. Please revise here and throughout the text.

Line 99: “Although AML arises in bone marrow hematopoietic stem cells, it quickly moves into the blood” This is misleading. AML blasts do not necessarily have to be found in the peripheral blood. Please revise.

Regarding survival of AML, an age-dependent consideration should be added as young patients, in particular paediatric patients, have a much better prognosis, even with high-risk features of AML.

Line 114: It should be written explicitly that AML1-ETO is a translocation (as indicated by t(8;21)). 

Line 209-211: A reference should be given. Futhermore, it is cytarabine, not cytarabine.

I disagree about the favourable toxicity profile of proteasome inhibitors in particular bortezomib. Serious toxicity can be observed with these agents This should be acknowledged in the text.

Authors should be spelled authors (with miniscule A) throughout the text.

Line 331: phosphorilated -> phosphorylated

Author Response

Reviewer 3

Dear Reviewer,

thank you very much for your kind revision of our work. Please find our point-by-point revision according to your suggestion.

We hope our manuscript is now suitable for publication in “Cancers”.

Sincerely,

Emiliano Fabiani

Comments and Suggestions for Authors

In their manuscript “The role of Forkhead box proteins in Acute Myeloid Leukemia”, Gurnari et al. discuss the role of FOX transcription factors for drug sensitivity in acute myeloid leukaemia (AML). 

Their introduction to FOX genes and proteins is concise, and their review provides a relevant summary.

Minor comments:

Line 59-62: “FOX proteins have been reported as active regulators of several networks, the main of which are: development, differentiation, maintenance of multipotency, proliferation, metabolism, DNA repair, cell cycle progression, migration, senescence, survival and apoptosis.” Relevant references should be incorporated.

According to your suggestion, the following references were added:

Golson, M.L.; Kaestner, K.H. Fox transcription factors: from development to disease. Development 2016, 143, 4558-4570.

He, L.; Gomes, A.P.; Wang, X.; Yoon, S.O.; Lee, G.; Nagiec, M.J.; Cho, S.; Chavez, A.; Islam, T.; Yu, Y.; et al. mTORC1 Promotes Metabolic Reprogramming by the Suppression of GSK3-Dependent Foxk1 Phosphorylation. Mol Cell 2018, 70, 949-960. 

Maachani, U.B.; Shankavaram, U.; Kramp, T.; Tofilon, P.; Camphausen, K.; Tandle, A.T. FOXM1 and STAT3 interaction confers radioresistance in glioblastoma cells. Oncotarget 2016, 7, 77365-77377. 

Alvarez-Fernández, M.; Medema, R.H. Novel functions of FoxM1: from molecular mechanisms to cancer therapy. Front Oncol 2013, 3, 30. 

Ackermann, S.; Kocak, H.; Hero, B.; Ehemann, V.; Kahlert, Y.; Oberthuer, A.; Roels, F.; Theißen, J.; Odenthal, M.; Berthold, F.; et al. FOXP1 inhibits cell growth and attenuates tumorigenicity of neuroblastoma. BMC Cancer 2014, 14, 840. 

Wang, P.; Lv, C.; Zhang, T.; Liu, J.; Yang, J.; Guan, F.; Hong, T. FOXQ1 regulates senescence-associated inflammation via activation of SIRT1 expression. Cell Death Dis 2017, 8, e2946. 

Macedo, J.C.; Vaz, S.; Bakker, B.; Ribeiro, R.; Bakker, P.L.; Escandell, J.M.; Ferreira, M.G.; Medema, R.; Foijer, F.; Logarinho, E. FoxM1 repression during human aging leads to mitotic decline and aneuploidy-driven full senescence. Nat Commun 2018, 9, 2834. 

Song, B.N.; Chu, I.S. A gene expression signature of FOXM1 predicts the prognosis of hepatocellular carcinoma. Exp Mol Med 2018, 50, e418. 

Gao, Y.F.; Zhu, T.; Mao, X.Y.; Mao, C.X.; Li, L.; Yin, J.Y.; Zhou, H.H.; Liu, Z.Q. Silencing of Forkhead box D1 inhibits proliferation and migration in glioma cells. Oncol Rep 2017, 37, 1196-1202. 

Line 96-97: “Acute myeloid leukaemia (AML) is a heterogeneous group of clonal disorder of the haematopoietic stem cells”. This is misleading as AML does not (necessarily) arise from the HSC. Please revise here and throughout the text.

We agree to your suggestion, the sentence was replaced in:

Acute myeloid leukemia (AML) is a heterogeneous group of clonal disorder of the hematopoietic compartment characterized by abnormal proliferation of undifferentiated myeloid progenitors, impaired hematopoiesis, bone marrow failure and variable response to therapy.

Line 99: “Although AML arises in bone marrow hematopoietic stem cells, it quickly moves into the blood” This is misleading. AML blasts do not necessarily have to be found in the peripheral blood. Please revise.

We agree to your suggestion, the sentence was replaced in:

Although AML arises in bone marrow hematopoietic stem cells, it may involve other extramedullary sites as lymph nodes, brain, spinal cord, liver, spleen, testicles and other parts of the body.

Regarding survival of AML, an age-dependent consideration should be added as young patients, in particular paediatric patients, have a much better prognosis, even with high-risk features of AML.

We agree to your suggestion, the following sentence was added in the text:

Incidence and survival outcomes vary according to age: in childhood it is a rare disorder with 7 occurrences per million annually and a 5-year overall survival (OS) exceeding 60%; in the adult setting AML is the most common form of acute leukemia with an incidence of 4-5 per 100.000 person/years and has the shortest survival (24% of 5 year-OS) [21-23].

21.    Zwaan, C.M.; Kolb, E.A.; Reinhardt, D.;, Abrahamsson, J.; Adachi, S.; Aplenc, R.; De Bont, E.S.; De Moerloose, B.; Dworzak, M.; Gibson, B.E.; et al. Collaborative Efforts Driving Progress in Pediatric Acute Myeloid Leukemia. J Clin Oncol 2015, 33, 2949-62. 

22.    Rubnitz, J.E. Current Management of Childhood Acute Myeloid Leukemia. Paediatr Drugs 2017, 19, 1-10. 

23.    Shallis, R.M.; Wang, R.; Davidoff, A.; Ma, X.; Zeidan, A.M. Epidemiology of acute myeloid leukemia: Recent progress and enduring challenges. Blood Rev 2019, 29. 

Line 114: It should be written explicitly that AML1-ETO is a translocation (as indicated by t(8;21)). 

We agree to your suggestion, the sentence was replaced in:

One of the most frequent initiating alteration in AML is the AML1-ETO translocation t(8;21) accounting for about 10% of total AML [25].

Line 209-211: A reference should be given. Futhermore, it is cytarabine, not cytarabine.

We agree to your suggestion,typing error was replacedand the following reference was added 

59. Quentmeier, H. , M. P. Martelli , and W. G. Dirks . et al. Cell line OCI/AML3 bears exon-12 NPM gene mutation-A and cytoplasmic expression of nucleophosmin. Leukemia 2005. 19:1760–1767.

I disagree about the favourable toxicity profile of proteasome inhibitors in particular bortezomib. Serious toxicity can be observed with these agents This should be acknowledged in the text.

Please find the following changes according to your suggestions:

Recently some data indicate that proteasome inhibitors suppress FOXM1 by inducing HSP70 [62]. Clinical trials have demonstrated their feasibility in combination with conventional chemotherapy for the treatment of resistant AML and acute lymphoblastic leukemia (ALL) with neurological toxicity as the most common adverse event [63]. Central and mostly peripheral neurotoxicity has been reported in case of bortezomib- the first in class proteasome inhibitor approved for the treatment of multiple myeloma- administration, resolved or controlled using neurotrophic drugs, gabapentin or neurophysiotherapy [64].

62.    Halasi, M.; Váraljai, R.; Benevolenskaya, E.; Gartel, A.L. A Novel Function of Molecular Chaperone HSP70: suppression of oncogenic FOXM1 after proteotoxic stress. J Biol Chem 2016, 291, 142-148.

63.    Attar, E.C.; Johnson, J.L.; Amrein, P.C.; Lozanski, G.; Wadleigh, M.; DeAngelo, D.J.; Kolitz, J.E.; Powell, B.L.; Voorhees, P.; Wang, E.S.; et al. Bortezomib added to daunorubicin and cytarabine during induction therapy and to intermediate-dose cytarabine for consolidation in patients with previously untreated acute myeloid leukemia age 60 to 75 years: CALGB (Alliance) study 10502. J Clin Oncol 2013, 31, 923–929.

64.    Bertaina, A.; Vinti, L.; Strocchio, L.; Gaspari, S.; Caruso, R.; Algeri, M.; Coletti, V.; Gurnari, C.; Romano, M.; Cefalo, M.G.; et al. The combination of bortezomib with chemotherapy to treat relapsed/refractory acute lymphoblastic leukaemia of childhood. Br J Haematol 2017, 176, 629-636. 

Authors should be spelled authors (with miniscule A) throughout the text.

Thank you for your suggestion, We leave the final decision to the Editor

Line 331: phosphorilated -> phosphorylated

Typing error was replaced according to your suggestion

Round 2

Reviewer 3 Report

All comments addressed.